# Virtual Feedback for Arm Motor Function Rehabilitation after Stroke: A Randomized Controlled Trial

**DOI:** 10.3390/healthcare10071175

**Published:** 2022-06-23

**Authors:** Silvia Salvalaggio, Pawel Kiper, Giorgia Pregnolato, Francesca Baldan, Michela Agostini, Lorenza Maistrello, Andrea Turolla

**Affiliations:** 1Laboratory of Rehabilitation Technologies, IRCCS San Camillo Hospital, 30126 Venice, Italy; giorgia.pregnolato@hsancamillo.it (G.P.); francescabaldan3@gmail.com (F.B.); lorenza.maistrello@hsancamillo.it (L.M.); 2Padova Neuroscience Center, Università degli Studi di Padova, 35131 Padova, Italy; 3IRCCS San Camillo Hospital, 30126 Venice, Italy; 4Department of Neuroscience, Section of Rehabilitation, University-General Hospital of Padova, 35122 Padova, Italy; michela.agostini@unipd.it; 5Department of Biomedical and Neuromotor Sciences—DIBINEM, Alma Mater Studiorum Università di Bologna, 40138 Bologna, Italy; andrea.turolla@unibo.it; 6Operative Unit Occupational Medicine, IRCCS Policlinico Sant’Orsola-Malpighi, 40138 Bologna, Italy

**Keywords:** stroke, rehabilitation, virtual reality, upper limb, motor learning

## Abstract

A single-blind randomized controlled trial was conducted to compare whether the continuous visualization of a virtual teacher, during virtual reality rehabilitation, is more effective than the same treatment provided without a virtual teacher visualization, for the recovery of arm motor function after stroke. Teacher and no-teacher groups received the same amount of virtual reality therapy (i.e., 1 h/d, 5 dd/w, 4 ww) and an additional hour of conventional therapy. In the teacher group, specific feedback (“virtual-teacher”) showing the correct kinematic to be emulated by the patient was always displayed online during exercises. In the no-teacher group patients performed the same exercises, without the virtual-teacher assistance. The primary outcome measure was Fugl-Meyer Upper Extremity after treatment. 124 patients were enrolled and randomized, 62 per group. No differences were observed between the groups, but the same number of patients (χ^2^ = 0.29, *p* = 0.59) responded to experimental and control interventions in each group. The results confirm that the manipulation of a single instant feedback does not provide clinical advantages over multimodal feedback for arm rehabilitation after stroke, but combining 40 h conventional therapy and virtual reality provides large effect of intervention (i.e., Cohen’s *d* 1.14 and 0.92 for the two groups, respectively).

## 1. Introduction

Impairments of the upper limb are the most disabling after stroke [1]. Among stroke survivors, 73% to 88% have acute hemiparesis [2]. Motor impairment are related to diverse aspects of movement, such as motor planning, execution, learning, and control [3]. Additionally, loss of sensation can contribute to motor control impairment due to inaccurate feedback affecting both task planning and voluntary motor output [4]. Plasticity in the motor cortex is more learning-dependent than use-dependent, since mere repetition of known movements does not induce neurophysiological and neuroanatomical changes, instead occurring when new motor skills are acquired [5,6]. Theoretical models propose different learning-paradigms (i.e., unsupervised, supervised, reinforced) [7] based on neuroplasticity, all mediated by a high number of movement repetitions, amplification of results, and multimodal augmented feedback (e.g., visual, verbal, haptic, auditory, kinematic) provided to subjects as coherently as possible with action execution [7,8,9]. Evidence on healthy subjects demonstrated that emulating the correct kinematic of a virtual teacher (augmented feedback), presented in a virtual environment, can promote learning of motor tasks better than training the same skill in a real environment, or in a virtual scenario without the virtual teacher. This evidence supports the hypothesis that reinforced learning-paradigms can enhance motor learning more than supervised and unsupervised learning-paradigms [10]. Virtual reality (VR) is a computer-based technology that is used as a therapeutic modality to provide an enriched training environment, aimed to improve motor learning by augmented feedback. Virtual tasks can be graded by clinicians for progressively challenging practice [7,11], with the advantage of being interesting, enjoyable, and motivating for patients [11]. In the last 15 years, the use of “reinforced feedback in virtual environment” for the recovery of upper limb after stroke, has been extensively investigated at the IRCCS San Camillo Hospital (Venice, Italy). A first longitudinal pilot study demonstrated that 20 h of intensive training induced significant changes in motor function and autonomy [12]. Then, the following controlled studies provided evidence that effects of VR were better than the same amount of conventional therapy (CT), both under experimental conditions [13] and in real clinical settings [14]. Recent evidence from extended meta-analyses in patients after stroke confirmed that VR is more effective than CT for the recovery of arm motor function measured by the Fugl-Meyer Assessment (FMA) scale, both as a single treatment and in addition to CT, when delivered for at least 15 h of intervention [15,16]. However, most of the factors influencing therapy outcome are still unknown, including the contextual factors (e.g., relational style of the therapist, expectations, mindset, and personality traits) composing the clinical setting [17,18]. To date, no studies have compared different VR settings displaying or not a virtual teacher as feedback. Based on these assumptions, we designed this clinical trial to transfer feasible theoretical models into clinical practice, to assess their potential relevance.

The purpose of the study was to assess whether the continuous visualization of a virtual teacher is effective in improving arm motor function in patients after stroke.

## 2. Materials and Methods

### 2.1. Trial Design and Population

A single-blind randomized controlled trial (RCT) was carried out on stroke patients with upper limb impairment, to compare the effects of displaying (i.e., Teacher group) or not (i.e., No-Teacher group) a continuous virtual teacher during a VR treatment. All consecutive inpatients admitted to the Stroke Rehabilitation Unit at the San Camillo IRCCS hospital in Venice (Italy) from April 2013 to March 2020 were considered eligible for the study. For better reporting of interventions, *CONsolidated Standards of Reporting Trials* (CONSORT) 2010 guidelines and its extension *CONSORT Statement for Randomized Trials of Nonpharmacologic Treatments* and *Template for Intervention Description and Replication* (TIDieR) checklist have been used [19,20,21]. All patients were informed on the aim of the study, and signed an informed consent form. Informed consent includes also procedures to withdraw at any time from the study, explicating that the decision will not affect the quality of health care received during hospitalization. Patients enrolled were randomly assigned to experimental or control groups according to a computer generated (Microsoft Excel) block random sequence. The allocation sequence was concealed from the principal investigator enrolling patients in sequentially numbered, opaque, and sealed envelopes. The researcher responsible for the randomization was independent of the assessors, assuring blindness to treatment allocation and randomization procedures.

Inclusion criteria for study population were: patients diagnosed with the first unilateral cortical-subcortical stroke, (ischemic or hemorrhagic); older than 18 years; a score between 0 (excluded) and 4 points (excluded) at the Italian version of the National Institutes of Health Stroke Scale [NIHSS-IT] upper limb subitem. Exclusion criteria were: bilateral or cerebellum infarcts; unstable medical conditions; fractures; major depressive disorders; traumatic brain injury; other neurological conditions; epilepsy; severe neglect; severe apraxia affecting evaluation; severe impairment of verbal comprehension hampering communication, not released informed consent, or lack of compliance.

### 2.2. The Virtual Reality Rehabilitation System

The Virtual Reality Rehabilitation System (VRRS^®^—Khymeia Group, Ltd., Noventa Padovana, Italy) was used to provide the virtual tasks. It consists of a computer connected to a 6-degrees of freedom electromagnetic motion tracking system (G4, Polhemus Ltd., Colchester, VT, USA) and a high-resolution screen displaying the virtual scenarios (Figure 1). The patient sits on a chair in front of the screen and wears a sensorized glove on the paretic arm (Figure 2). The glove does not provide haptic feedback, but only contains a motion tracking sensor, matched to the virtual object representing the end-effector of the biomechanical system. The virtual object is displayed in a shape similar to the real one and its displacement is synchronized in time and space between real and virtual environments.

### 2.3. Assessment

To compare the clinical and kinematics efficacy of different interventions, both clinical and instrumental outcomes were assessed before (T0) and after (T1) treatment. These assessments were carried out by clinicians from the research department, all blind to patients’ allocation. The primary outcome was the Fugl-Meyer Assessment Upper Extremity (FMA-UE) [22] for arm motor function. Other clinical outcome measures were as follows: Fugl-Meyer Assessment for sensory function (Fugl-Meyer sensation, FMA-sensation) and range of motion and pain (Fugl-Meyer pain/rom, FMA-pain/rom) [22]; Nine-Hole Pegboard Test (NHPT) [23] and Box & Blocks Test (BBT) for manual dexterity; Reaching Performance Scale (RPS) [24] for reaching ability; [25]; Modified Ashworth Scale (MAS) for measuring muscle tone in pectoralis major, biceps brachii, flexor carpi, flexor digitorum profundus, flexor digitorum superficialis [26]; and Functional Independence Measure (FIM) [27] for autonomy in activities of daily living (ADLs). The instrumental evaluation consisted of eight exercises involving different arm movements: (1) Tight—slide (Upper Limb front sliding); (2) Elbow flexion extension; (3) Contralateral hip—ipsilateral knee; (4) Simple reaching; (5) Forearm pronation—supination; (6) Shoulder abduction with elbow flexion; (7) Shoulder elevation; and (8) Shoulder abduction with external rotation and 90° elbow flexion. For each exercise, the patient was asked to perform 10 repetitions and no physical assistance was provided by the physiotherapist. Kinematic outcome measures of the 8 movements considered were: mean duration (in seconds), that is the time needed to complete a single movement; mean linear velocity (speed, in mm/s); mean number of sub-movements (in No.), intended as number of peaks of acceleration during movement execution.

### 2.4. Intervention

Patients in both groups (i.e., Teacher group, No-Teacher group) received the same amount of VR therapy (i.e., 1 h/d, 5 d/w, for 4 weeks: 20 h overall) and an additional hour of CT per day, for an amount of 40 h of treatment overall, in 4 weeks. In the case of missed sessions, they were rescheduled during hospitalization to complete the assigned rehabilitation program. Each session was delivered individually.

#### 2.4.1. Conventional Therapy

The CT treatment consisted of whole-body exercises that were selected autonomously by the clinician and performed in a gym or a private room for 1 h/d. Each session was adapted to the patient’s clinical condition and motor ability to perform exercises, avoiding any harm that may occur (e.g., patients’ referring shoulder pain). In particular for the arm, patients were asked to perform exercises including shoulder and elbow flexion–extension, shoulder abduction–adduction, internal–external rotation, circumduction, and forearm pronation-supination.

#### 2.4.2. Virtual Reality Rehabilitation

Each VR rehabilitation session was individually delivered to the patients in a dedicated room at the Laboratory of Rehabilitation Technologies, for 1 h a day. The patient was sitting on a standard chair in front of the screen, wearing the sensorized glove on the paretic arm. The physiotherapist was present during the rehabilitation session monitoring the VR environment and patient performance. From a technical perspective, the exercise in the virtual environment was the kinematic of a point-to-point task pre-recorded by the therapist moving her/his sensorized end-effector according to an ideal path, in the free space. This recorded trajectory was displayed to the patient and represented the exercise to be performed. Each trial of an exercise was self-paced by the patient by a parser in both the start and end zones of the task, initiating and stopping the repetition, respectively. To allow the therapist to adapt task difficulty to patient motor function, the trial can also be stopped manually, to start a new one. The “virtual teacher” was the animation of the virtual object moved by the therapist, which automatically executed the correct trajectory. In the ‘teacher’ condition, the ideal and the patient end-effectors were superimposed but with different colors, allowing the patient to emulate a model performing the trajectory without spatial errors for that kinematic. This emulation was considered as a translated paradigm of supervised learning, in the real clinical setting. Thus, the teacher end-effector was always displayed on the screen, but its animation started only after the patient entered in the start zone, then moved to the end-zone according to the real velocity of the original execution, staying there till the end of the task defined by the patient entering in the end-zone. Given this, the teacher showing the correct path to follow (and potentially to be emulated) was considered as feedback, since the patient voluntarily activating the start of the predetermined information was always displayed identically. In the Teacher group, the “virtual teacher” feedback was displayed online during each task repetition, whereas, in the No-Teacher group, the patient was asked to perform the same motor exercises using the upper limb, but without the enhanced visual feedback provided via the virtual teacher (Figure 3).

Virtual tasks consisted of both simple movements, involving a single joint of the upper limb (e.g., moving a ball by executing elbow flexion in vertical plane) and complex movements that involved multiple muscle synergies (e.g., putting a glass on a shelf, using a toothbrush). All exercises were tailored to the patient’s condition and arm motor function. After each trial, acoustic feedback provided information at the end of repetitions. After each single repetition, a score proportional to the amount of spatial error was provided by means of a numeric value and its graphic representation through a bar graph. The numerical value, expressed on a scale from 0 to 100, gives an indication of the quality of the movement (i.e., 0 = “movement not executed or completed” and 100 “movement correctly executed”), defined as the amount of spatial error expressed with regard to the ideal trajectory. The physiotherapist provided verbal feedback on movement quality or compensatory strategies. The frequency of verbal feedback was not standardized nor different between groups, but provided according to the clinical judgement of each physiotherapist.

### 2.5. Sample Size Calculation

Sample size calculation was performed on the basis of the available evidence on FMA-UE as primary outcome and an effect size of d = 0.54 was considered large enough to detect a significant difference between groups in upper limb function [13,23]. Power calculation indicates that a sample size of 55 patients per group, given α = 0.05 and 1 − β = 0.8, would be sufficient to observe a significant difference between groups. Nonetheless, considering a potential drop-out rate of 10% (i.e., 6 patients per group) the final sample size was 61 per group (i.e., 122 patients overall).

### 2.6. Statistical Analysis

Both descriptive and inferential analyses were performed for both groups separately. Parametric and non-parametric tests were chosen according to distribution shape, evaluated by the Shapiro–Wilk test. Differences between groups at baseline were investigated using χ^2^ test for frequencies and unpaired samples t-Student or Wilcoxon Mann–Whitney tests for the other variables. According to data distribution, differences in clinical outcome measures after treatment within and between groups were compared using parametric tests (i.e., paired and unpaired *t*-test) or nonparametric tests (Wilcoxon test and Wilcoxon Mann–Whitney test). For each outcome measure, effect sizes were calculated with Cohen’s *d* [28]. Subsequently, patients were stratified according to stroke etiology, time from lesion, and responsiveness to therapy, defined as an improvement greater than the minimally clinically important difference (MCID) or the minimal detectable change (MDC) at clinical outcomes, only if available in the literature. For responsiveness stratification, MCID was considered for FMA-UE (5 points) [29] and FIM (22 points) [30], while MDC for RPS (4 points) [31] and NHPT (54%) [32]. Linear regression models were used to infer any potential relation between clinical and kinematics outcomes after treatment (i.e., dependent variables) and baseline clinical or demographic characteristics (i.e., independent variables), in the whole sample, treatment groups and subgroups. Only those models with a determination coefficient (high percentages of McFadden’s R2) higher than 0.8 (high goodness-of-fit of the model) were considered for reporting. To perform an intention-to-treat (ITT) analysis, all the outcomes were assessed for all the enrolled patients independently of their adherence to the treatment. For ITT approach, baseline values were considered also as treatment results in case of missing data. All statistical analyses were performed using the free software RStudio Team (2016) [33]. Statistical significance was set at *p*-value ≤ 0.05.

## 3. Results

Among the whole cohort of stroke survivors regularly accepted at the San Camillo IRCCS hospital for rehabilitation needs, approximately 40% of them are potential candidates for upper limb VR treatment, because of the IT-NIHSS criterium. From 2013 to 2020, 450 patients were accepted at San Camillo IRCCS Hospital; among them, 180 accomplished the IT-NIHSS criterium, and thus 124 were randomized due to the other criteria. The complete flow of the trial is displayed in Figure 4.

Demographic (Table 1) characteristics were comparable between groups after allocation, since all *p*-values of Chi-Squared test and *t*-test were *p* > 0.05.

Both groups showed statistically significant improvement from baseline to follow-up, except for MAS at biceps brachii in the No-Teacher group (*p* = 0.84), but no differences between groups were observed (Table 2).

After treatment, 68 patients overall improved over the MCID in the FMA-UE (responders). The number of responders to the VR therapy in the Teacher (36 out of 62, 58%) and No-Teacher groups (32 out of 62, 52%) was comparable (χ^2^ = 0.29, *p* = 0.6), as were their demographic characteristics and improvement at FMA-UE, FMA-sensation, FMA-pain/ROM, RPS, NHPT, and FIM within and between groups. Only the MAS biceps brachii did not improve in both groups. Regression models of the overall sample showed that improvement in FMA-UE was significantly influenced by baseline values of upper limb motor function (FMA-UE), sensation (FMA sensation), independence (FIM), and spasticity at the biceps brachii (MAS biceps brachii). Similar influences were also observed for improvement at the FMA-UE and FIM in the No-Teacher and the Non-Responders in the FMA-UE groups. Conversely, improvement at the FMA-UE was only influenced by the FMA-UE at baseline both in the Teacher and the Responder at the FMA-UE groups. Finally, improvement in the RPS was significantly influenced by baseline values of the RPS and FMA-UE, regardless of whether patients reached or not the MCID in the RPS. Overall, regression analysis confirmed that belonging to a treatment group did not influence motor recovery (Table 3).

All the Responders at the FMA-UE, regardless their treatment group allocation (*n* = 68), improved significantly more than Non-Responders patients (*n* = 56) in all the outcome measures, except for MAS in the biceps brachii (Table 4).

From the kinematic perspective, each exercise improved significantly from baseline to the end of treatment, but no differences were found between the Teacher and No-Teacher groups. Neither regression analyses found significant influence of clinical and demographic characteristics on the kinematic outcome (R^2^ lower than 0.8 for all the models). Finally, none of the patients experienced adverse events or side effects, such as cybersickness, nausea, or headache, which may occur when interacting with a virtual reality environment.

## 4. Discussion

Todorov et al. [10] demonstrated in healthy subjects that during the acquisition of a new visuomotor task, augmented feedback in VR promotes better motor learning than exercising in a real environment or in an unsupervised condition. Clinical evidence on stroke rehabilitation does not recommend providing only autonomous exercise practice, thus the constant presence of a physical therapist is required. For this reason, we designed our clinical trial to compare different conditions in a virtual scenario providing or not augmented feedback promoting supervised learning. Results showed that the two groups were homogeneous and comparable at baseline, both for age and months after stroke (*p* > 0.05 for *t*-test) and for sex, type of stroke, and affected hemisphere (χ^2^ test, *p* > 0.05), ensuring that improvements achieved were not influenced by allocation. Moreover, the total dose of CT and VR rehabilitation showed a positive effect on upper limb recovery, regardless of whether it displayed a virtual teacher, as shown by the large effect size for the primary outcome measure (i.e., 1.14 for the Teacher group and 0.92 for the No-Teacher group). This is also confirmed by the regression models where the treatment group did not influence the recovery significantly.

### 4.1. Neuromotor Interpretation

The main finding of this clinical trial (comparable results between displaying or not just one pattern of movements) has two potential interpretations. On the one hand, the presentation of different amounts of concurrent virtual feedback provided simultaneously during motor action does not improve motor learning after stroke. In fact, it is arguable that the constant presence of the physiotherapist acts as the main cue promoting the motor outcome improvement, since visual, auditory, and verbal feedback are continuously provided to patients in combination with those from the virtual environment. All together they represent a unique enriched environment relevant for stimulating neuroplasticity and motor learning, even if the virtual teacher is not displayed. On the other hand, these results support the finding that rather than single planned movement, the nervous system has an abundance of different solutions made up of a very high number of kinematic combinations from which movements can be produced to achieve the same goals [34]. In this regard, this evidence supports the dynamical perspectives of motor control [35].

### 4.2. Clinical and Pragmatic Interpretation

From a clinical perspective, more than half of the patients improved more than the MCID at the FMA-UE (5 points) [29] in each group, confirming that the combination of CT and VR rehabilitation has both statistical and clinical important effects, regardless of the presence or not of specific feedback (the virtual teacher in our trial). Conversely, a lack of response to therapy seems to be associated with spasticity at the flexor carpi muscle. The results of this study support the evidence from several trials on the positive effects induced by VR rehabilitation after stroke [1,13,14,15]. However, it should be acknowledged that the patients enrolled in the study were also undergoing also CT during the same period of VR rehabilitation, so the positive effects observed in the motor domain could also be linked to the high dose of therapy delivered and not only to the rehabilitation modalities received. In this regard, to better test transferability of theoretical models of motor learning to clinical practice, an adapted intervention protocol based on higher intensity of movement repetitions along shorter exposure time might be a better experimental design to investigate potential effects of single feedback on motor learning. Another limitation of the study is that long-term follow-up was not available for these patients, although it is unlikely that a difference between groups would emerge later, since no evidence is present yet in the literature in favor of such an effect of a single rehabilitation modality. Patients were mainly in the chronic phase after stroke (7 months from stroke onset on average), when motor improvement is harder to detect in such therapy settings and this may have contributed to the lack of difference between groups, despite the large number of participants. Future research might promote multicenter studies in the field, both to speed up the enrolment rate and to improve the external validity of using VR for the recovery upper limb after stroke. A low level of spasticity and a high residual motor function appear to be positive prognostic factors to induce more independence in ADLs and better motor recovery after VR training in stroke survivors.

## 5. Conclusions

The clinical messages of this study are that providing visual teacher feedback during a movement task in an enriched environment does not alter motor learning, and the manipulation of only one type of feedback (i.e., the presence visualization of a virtual teacher) is not sufficient, in a clinical setting already providing multimodal stimulations, to induce significant differences for direct exploitation of supervised and reinforced paradigms of motor learning. However, adding VR to CT provides a large intervention effect.

Future research should address the effect of different combinations of feedback (e.g., visual, auditory, virtual teacher, or haptic) provided to patients during virtual reality rehabilitation. Thus, clinical trials should be designed to compare different kinds of feedback to low/no feedback to patients, instead of small differences in similar settings of multimodal ones, with the aim of investigating the real effect of cumulative feedback in real clinical practice for promoting motor learning in stroke rehabilitation.

## Figures and Tables

**Figure 1 healthcare-10-01175-f001:**
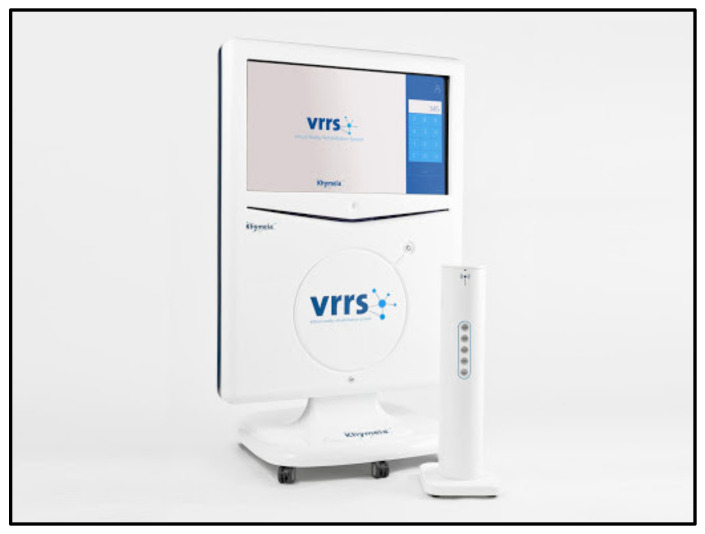
The Virtual Reality Rehabilitation System (VRRS^®^).

**Figure 2 healthcare-10-01175-f002:**
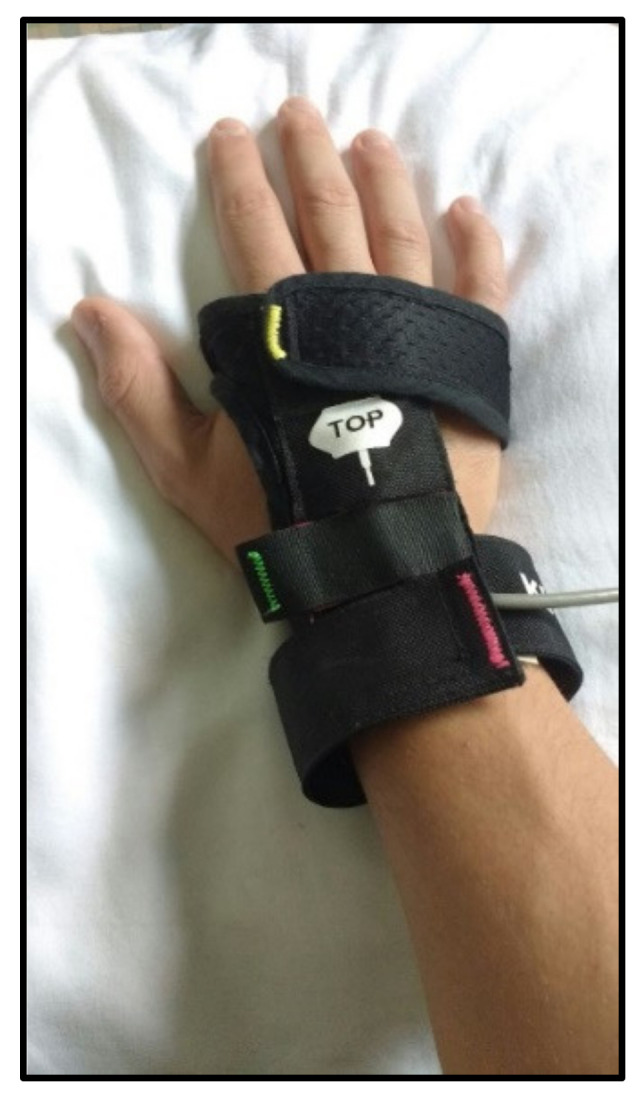
Sensorized electromagnetic glove.

**Figure 3 healthcare-10-01175-f003:**
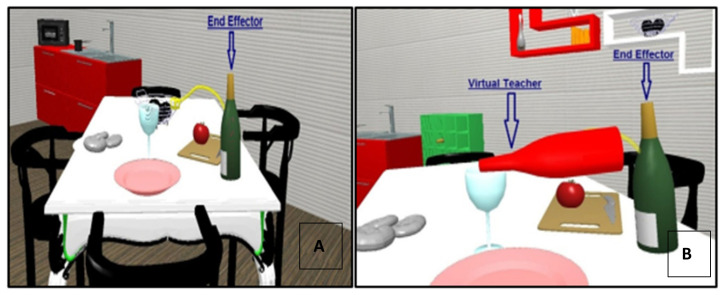
Virtual scenarios of the Virtual Reality Rehabilitation System (VRRS^®^). (**A**) On the left, the scenario presented to the No-Teacher group: the green bottle is the end-effector which is controlled by patient’s hand wearing the sensorized glove. The task to accomplish is represented by the exact trajectory to accomplish, but no virtual teacher can be emulated to perform the task accurately. (**B**) On the right, the scenario presented to the Teacher group: the red bottle represents the virtual teacher, performing the exact trajectory displayed by the Virtual Reality Rehabilitation System (VRRS). In real-time the patient can emulate with the green bottle (controlled by own movements) the exact movement of the virtual teacher (red bottle).

**Figure 4 healthcare-10-01175-f004:**
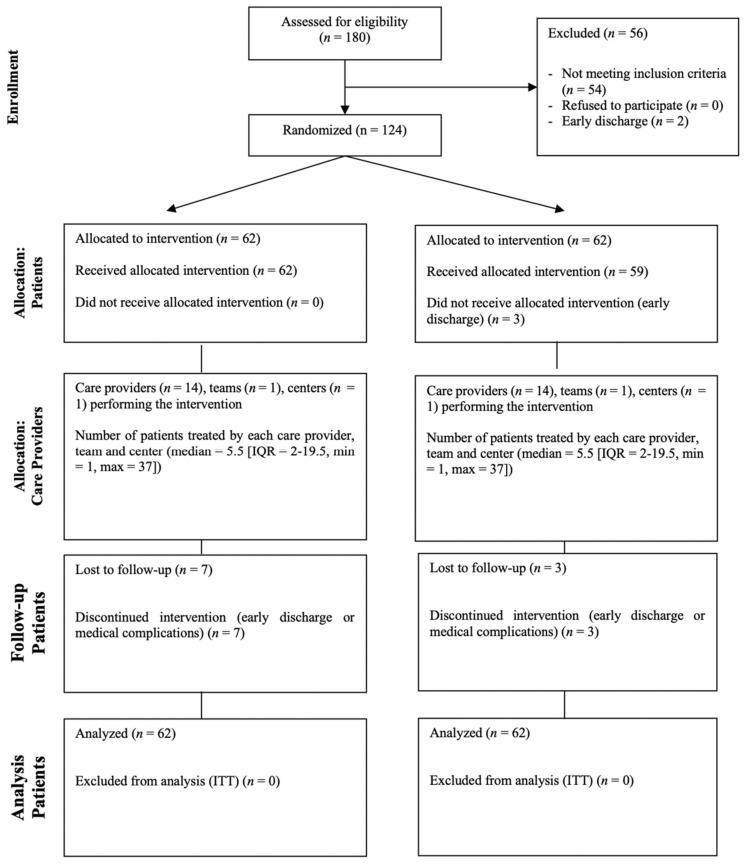
Modified CONSORT flow diagram for individual randomized controlled trials of nonpharmacologic treatments. IQR = interquartile range; max = maximum; min = minimum; *n* = number of subjects.

**Table 1 healthcare-10-01175-t001:** Characteristics of the teacher, and no-teacher groups at baseline.

Demographics	Overall (*n* = 124)	Teacher Group (*n* = 62)	No-Teacher Group (*n* = 62)	*p*-Value
Sex Total Male (%)/Female (%)	79 (64%)/ 45 (36%)	42 (68%)/ 20 (32%)	37 (60%)/ 25 (40%)	*p* = 0.4
Age Mean ± SD/Median (range)	62.58 ± 14.01/ 64 (19–92)	63.944 ± 13.58/ 66 (24–84)	61.21 ± 15.12/ 62 (19–84)	*p* = 0.3
Type of stroke Ischemic (%)/Hemorrhagic (%)	93 (75%)/ 31 (25%)	49 (79%)/ 13 (21%)	44 (71%)/ 18 (29%)	*p* = 0.3
Hemisphere Right (%)/Left (%)	65 (52%)/ 59 (48%)	34 (55%)/ 28 (45%)	31 (50%)/ 31(50%)	*p* = 0.6
Months from injury Mean ± SD/Median (range)	7.36 ± 14.65/ 3 (1–144)	5.71 ± 8.69/ 3 (1–60)	9.01 ± 18.78/ 4 (1–144)	*p* = 0.2

**Caption.** Values are reported as number and percentages, mean ± standard deviation (sd), median (range). Between analysis of demographic characteristics at baseline of the two groups of patients are reported to investigate whether the two groups were comparable at baseline. *t*-test was used for age and months from injury, χ^2^ test was used for sex, type of stroke, and hemisphere. Statistical significance was set at *p*-value < 0.05. The results show that the two groups were equal for each demographic characteristics at baseline, therefore comparable.

**Table 2 healthcare-10-01175-t002:** Descriptive and inferential analysis of teacher and no-teacher groups at baseline (T0) and after treatment (T1).

Outcome Measures	Teacher (*n* = 62)				No Teacher (*n* = 62)				Between Group
	T0	T1	Within Group	Effect Size	T0	T1	Within Group	Effect Size	
**FMA-UE** Mean ± SD/Median (range)	40.77 ± 14.27 43 (14–62)	47.06 ± 14 50 (17–66)	*p* < 0.01 *	1.14	36.45 ± 14.27 36 (0–62)	41.97 ± 15.61 43 (13–66)	*p* < 0.01 *	0.92	*p* = 0.4
**FMA-sensation** Mean ± SD/Median (range)	18.42 ± 6.84 21 (0–24)	20.06 ± 5.75 23 (4–24)	*p* < 0.01 *	0.42	18.53 ± 6.25 21 (0–24)	20.66 ± 5 22 (0–24)	*p* < 0.01 *	0.53	*p* = 0.3
**FMA-pain/rom** Mean ± SD/Median (range)	42.08 ± 7.04 44 (17–48)	43.58 ± 5 45 (17–48)	*p* < 0.01 *	0.31	41.73 ± 5.29 43 (27–48)	43.03 ± 4 43 (30–48)	*p* < 0.01 *	0.36	*p* = 0.5
**RPS** Mean ± SD/Median (range)	23.66 ± 11 28 (0–36)	28.1 ± 9 32 (0–36)	*p* < 0.01 *	0.83	20.6 ± 11.23 22 (0–36)	24.08 ± 10 24 (4–36)	*p* < 0.01 *	0.57	*p* = 0.2
**NHPT** Mean ± SD/Median (range)	0.12 ± 0.3 0.02 (0–2)	0.15 ± 0.2 0.06 (0–0.56)	*p* < 0.01 *	0.08	0.12 ± 0.2 0 (0–0.76)	0.16 ± 0.2 0.01 (0–0.75)	*p* < 0.01 *	0.27	*p* = 0.4
**FIM** Mean ± SD/Median (range)	93.32 ± 20.79 100 (38–123)	103 ± 18 100 (64–126)	*p* < 0.01 *	0.91	91.56 ± 21.89 96 (33–123)	99.10 ± 21.69 103 (42–126)	*p* < 0.01 *	0.73	*p* = 0.7
**MAS biceps brachii** Mean ± SD/Median (range)	0.78 ± 1 1 (0–4)	0.58 ± 0.86 0 (0–4)	*p* < 0.01 *	0.30	0.84 ± 0.94 1 (0–4)	0.76 ± 0.88 1 (0–3)	*p* = 0.84	0.10	*p* = 0.2

**Caption.** FIM: Functional Independence Measure; FMA-pain/rom: Fugl-Meyer Assessment Scale pain and range of motion subitem; FMA sensation: Fugl-Meyer Assessment Scale sensation subitem; FMA-UE: Fugl-Meyer Assessment Scale Upper Extremity subitem; MAS biceps brachii: Modified Ashworth Scale at the biceps brachii; RPS: Reaching Performance Scale; T0: before treatment; T1: after treatment. Values are reported as mean ± standard deviation (sd), median (range). Wilcoxon-Mann–Whitney U test was used for between group analysis to investigate whether the intervention in the Teacher group was more effective than the intervention in the No-Teacher group Wilcoxon signed-rank test was used for within group analysis to investigate whether each intervention provided a significant improvement in both groups. Statistical significance was set at *p*-value < 0.05. (*). Cohen’s *d* was used to calculate effect size of the two groups, showing the effect of intervention detected by each outcome measure (Cohen’s *d* > 0.5 large effect, 0.5–0.2 medium effect, 0.1 small effect).

**Table 3 healthcare-10-01175-t003:** Generalized Linear Model with good variability explained, for the overall sample and subgroups.

**Overall Population (*n* = 124)**			
**Outcome**	**Model**	**R^2^**	**Residuals S-W test**
ΔFMA-UE	25.54 + 0.13 FMA-UE T0 + 0.30 FMA sensation T0 + 0.72 FIM T0—2.44 MAS biceps brachii T0	0.80	*p* = 0.02
**Teacher group (*n* = 62)**			
**Outcome**	**Model**	**R^2^**	**Residuals S-W test**
FMA-UE T1	10.72 + 0.89 FMA-UE T0	0.85	*p* = 0.03
**No-Teacher group (*n* = 62)**			
**Outcome**	**Model**	**R^2^**	**Residuals S-W test**
ΔFMA-UE	13.62 + 0.3 FMA-UE T0 + 0.82 FIM T0	0.81	*p* = 0.3
FIM T1	13.62 + 0.30 FMA-UE T0 + 0.82 FIM T0	0.81	*p* = 0.32
**Responders FMA-UE (*n* = 68)**			
**Outcome**	**Model**	**R^2^**	**Residuals S-W test**
FMA-UE T1	13.80 + 0.91 FMA-UE T0	0.91	*p* = 0.02
**Non-Responders FMA-UE (*n* = 56)**			
**Outcome**	**Model**	**R^2^**	**Residuals S-W test**
FMA-UE T1	1.07 + 1.72 sex M + 0.98 FMA-UE T0 -0.68 MAS flexor carpi T0	0.99	*p* = 0.08
FIM T1	13.00 + 4.89 sex M + 0.15 FMA-UE T0 + 0.33 FMA sensation T0 + 0.76 FIM T0	0.88	*p* = 0.82
**Responders RPS (*n* = 43)**			
**Outcome**	**Model**	**R^2^**	**Residuals S-W test**
RPS T1	12.37 + 0.91 FMA-UE T0—0.77 MAS tot T0	0.80	*p* = 1
**Non-responders RPS (*n* = 81)**			
**Outcome**	**Model**	**R^2^**	**Residuals S-W test**
RPS T1	10.84—0.09 age + 2.52 sex M + 0.62 FMA-UE T0 + 0.58 RPS T0	0.91	*p* = 1

**Caption.** FIM: Functional Independence Measure; FMA sensation: Fugl-Meyer Assessment Scale sensation subitem; FMA-UE: Fugl-Meyer Assessment Scale Upper Extremity subitem; ΔFMA-UE: mean improvement at the FMA-UE; M: male; MAS biceps brachii: Modified Ashworth Scale at the biceps brachii; MAS flexor carpi: Modified Ashworth Scale at the flexor carpi; MAS tot: Modified Ashworth Scale total; RPS: Reaching Performance Scale; T0: before treatment; T1: after treatment. R2: High percentages of McFadden’s R^2^, Residuals S-W test: residuals at the Shapiro–Wilk test; (*p*-value > 0.05 at the residuals indicate goodness-of-fit for the models). The independent variables described in the models represent the clinical features significantly predicting the response outcome.

**Table 4 healthcare-10-01175-t004:** Clinical results for the Responder and Non-Responder groups before and after treatment.

Outcome Measures	Responder (*n* = 68)			Non-Responder (*n* = 56)			Between Groups
	T0	T1	Within Group	T0	T1	Within Group	
**FMA-UE** Mean ± SD/ Median (range)	39 ± 14 39 (8–60)	49 ± 13 50 (15–66)	*p* < 0.01 *	38 ± 15 36 (13–62)	39 ± 15 38 (13–65)	*p* = 0.01 *	*p* < 0.01 *
**FMA-sensation** Mean ± SD/ Median (range)	19 ± 6 21 (0–24)	21 ± 4 24 (7–24)	*p* < 0.01 *	18 ± 7 21 (0–24)	19 ± 5 22 (0–24)	*p* = 0.01 *	*p* < 0.01 *
**FMA-pain/rom** Mean ± SD/ Median (range)	42 ± 6 44 (17–48)	44 ± 4 45 (30–48)	*p* < 0.01 *	42 ± 6 43 (17–48)	42 ± 5 43 (17–48)	*p* = 0.3	*p* = 0.02 *
**RPS** Mean ± SD/ Median (range)	24 ± 11 28 (0–36)	29 ± 10 32 (4–36)	*p* < 0.01 *	20 ± 11 21 (0–36)	23 ± 10 21 (0–36)	*p* < 0.01 *	*p* = 0.03 *
**NHPT** Mean ± SD/ Median (range)	0.12 ± 0.23 0.08 (1–0.5)	0.22 ± 0.19 0.18 (0–0.75)	*p* < 0.01 *	0.11 ± 0.29 0 (0–2)	0.08 ± 0.19 0 (0–0.41)	*p* = 0.3	*p* < 0.01 *
**FIM** Mean ± SD/ Median (range)	92 ± 22 99 (33–123)	103 ± 20 112 (42–126)	*p* < 0.01 *	93 ± 21 96 (38–123)	98 ± 19 100 (51–124)	*p* < 0.01 *	*p* < 0.01 *
**MAS biceps brachii** Mean ± SD/ Median (range)	1 ± 1 0 (0–3)	1 ± 1 0 (0–3)	*p* = 0.09	1 ± 1 1 (0–4)	1 ± 1 1 (0–4)	*p* = 0.2	*p* = 0.8

**Caption.** FIM: Functional Independence Measure; FMA-pain/rom: Fugl-Meyer Assessment Scale pain and range of motion subitem; FMA sensation: Fugl-Meyer Assessment Scale sensation subitem; FMA-UE: Fugl-Meyer Assessment Scale Upper Extremity subitem; MAS biceps brachii: Modified Ashworth Scale at the biceps brachii; RPS: Reaching Performance Scale; T0: before treatment; T1: after treatment. Values are reported as mean ± standard deviation (sd), median (range). Wilcoxon-Mann–Whitney U test was used for between group analysis to investigate whether the intervention in the Responder group was more effective than the intervention in the Non-Responder group, *t*-test or Wilcoxon signed-rank test was used for within group analysis to investigate whether each intervention provided a significant improvement in both groups. Statistical significance was set at *p*-value < 0.05. (*) Results show that Responders had a clinical response higher than Non-Responders.

## Data Availability

The datasets presented in this article are not readily available because the authors do not have permission to share raw data. The results produced in the present study are included in the article. Requests to access the datasets should be directed to silvia.salvalaggio@hsancamillo.it.

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
