# Peer review of "Virtual Feedback for Arm Motor Function Rehabilitation after Stroke: A Randomized Controlled Trial"

_healthcare, 2022, doi:10.3390/healthcare10071175_

Round 1

Reviewer 1 Report

Major comments and suggestions:

1.      The process introduced in Section 2.3 and in the caption of Figure 3 need to be reorganized in a clearer manner. Squeezing all the steps in one giant paragraph is not reader friendly.

2.      The parameters in Sections 2.4 and 2.5 need to be explained.

3.      Please reorganize the manuscript so that one table can be on the same page.

4.      Are the “R2’s” in Table 3 actually “R2’s”? If so, please format them correctly.

5.      The tables are not informative in the current states. Some visualization of the data should be provided.

6.      There are a lot of statistical methods applied in this manuscript, while the main conclusion is based on the R2 values of regression analyses. Either the data analyses are poorly described, or some important points are drowned in the flood of all kinds of jargons.

7.      It is difficult to find the connection between and discussion in Section 4 and the data analyses in Section 3. A paragraph occupying an entire page is unpleasant to read, either.

Author Response

Response to reviewer 1

Major comments and suggestions:

  1. The process introduced in Section 2.3 and in the caption of Figure 3 need to be reorganized in a clearer manner. Squeezing all the steps in one giant paragraph is not reader friendly.

The requested changes have been made according to the reviewer’s suggestion. Section 2.3 has been divided into two different sections for the description of Conventional Therapy (CT) and Virtual Reality (VR) rehabilitation. Moreover, caption of Figure 3 has been reorganized for a more reader friendly description.

  1. The parameters in Sections 2.4 and 2.5 need to be explained.

The parameters in section 2.4 are Cohen’s D, for defining the effect size, that is the level of the effect of the treatment. α (Alpha) is the probability of Type I error in any hypothesis test-incorrectly rejecting the null hypothesis, and 1-β (Beta) is the power of the study, derived from β which is the probability of making a Type II error or false negative rate. The parameter in section 2.5 is McFadden's R2, which estimates the ability of the regression model to predict correctly the investigated condition, then the higher is the value the better is the goodness-of-fit of the model. We have reported the information in the statistical section according to the standard reporting of sample size calculation provided by the CONSORT guidelines, which has also been revised for a double check by a professional statistician, member of the authors team. The description of the clinical and kinematic outcome measured used for the assessment have been described in the section 2.3.   

  1. Please reorganize the manuscript so that one table can be on the same page.

Thanks to the reviewer for the suggestion. The required change has been made for all the tables of the manuscript.

  1. Are the “R2’s” in Table 3 actually “R2’s”? If so, please format them correctly.

Yes, the R2 is actually R2, the format has been corrected in the Table 3. Thanks to this suggestion, Table 3 has been improved by adding into the table the description of which groups corresponds to which model.

  1. The tables are not informative in the current states. Some visualization of the data should be provided.

Thanks to the reviewer for the suggestion. Unfortunately, for the kind of data and results we have, graphs would not be very informative. The authors therefore decided to keep the tables, also to make the exact data more accessible to future secondary research studies.

  1. There are a lot of statistical methods applied in this manuscript, while the main conclusion is based on the R2values of regression analyses. Either the data analyses are poorly described, or some important points are drowned in the flood of all kinds of jargons.

Thanks to the reviewer for the suggestion. Calculation of the effect size for both groups was added in Table 2, considered as an important information regarding the effect of the intervention proposed in this trial. More information related to results of Table 1 and 2 have expanded in the discussion and conclusion sections. Data analysis description has been improved by more details of both the statistical analysis section in the methods and the Table 2 caption.

  1. It is difficult to find the connection between discussion in Section 4 and the data analyses in Section 3. A paragraph occupying an entire page is unpleasant to read, either.

Thanks to the reviewer for the suggestion. A more readable description of section 4 has been improved, trying to add more logical connections between the data analysis and discussion. Moreover, section 4 has been divided into two paragraphs for a more reader-friendly section.

Reviewer 2 Report

Review of manuscript ID: healthcare-1741023

The paper presents interesting research on virtual reality and variants of using it in the treatment of patients after stroke.

The title encourages you to read the content of the article.

The summary is complete and meets the requirements.

The purpose of the publication was clearly defined.

Presentation methods are appropriate, although in this section, material, and method, in my opinion, there is no explanation of what the work with the patient was about apart from the use of virtual reality (verse 145 - 146).

The authors point out in the introduction to the work that damage to the upper limb occurs just after a stroke. Their aim was to assess the impact of virtual reality, and in particular to strengthen the therapy by adding a virtual teacher, but the effects of therapy for the upper limb depend not only on the therapy of the only affected upper limb and also on other parts of the body, especially the trunk. Properly selected functional therapy can, among other things, normalize the muscle tension of patients after a stroke, and as they write in their conclusions, the results depended on the level of muscle tension. The more I believe that the section material and method should be supplemented with relevant information.

Moreover, please analyze the flow diagram. It is not clear why each group of 62 patients was analyzed if several people in each group were either uninterrupted or/and did not continue to intervene.

The work is interesting and possible to publish after completion/correction.

Author Response

Response to Reviewer 2

  1. Presentation methods are appropriate, although in this section, material, and method, in my opinion, there is no explanation of what the work with the patient was about apart from the use of virtual reality (verse 145 - 146).

Thanks to the reviewer for the comment, allowing us to describe the intervention more clearly by dividing the intervention for a full description of the Conventional Therapy (CT) and Virtual Reality (VR) intervention. Indeed, the upper limb rehabilitation program of this trial consisted of both CT in the gym and VR in a dedicated room for the virtual scenario, as described in sections 2.4.1 and 2.4.2.

  1. The authors point out in the introduction to the work that damage to the upper limb occurs just after a stroke. Their aim was to assess the impact of virtual reality, and in particular to strengthen the therapy by adding a virtual teacher, but the effects of therapy for the upper limb depend not only on the therapy of the only affected upper limb and also on other parts of the body, especially the trunk. Properly selected functional therapy can, among other things, normalize the muscle tension of patients after a stroke, and as they write in their conclusions, the results depended on the level of muscle tension. The more I believe that the section material and method should be supplemented with relevant information.

Thanks to the reviewer for the suggestion, the authors agree that higher level of muscle tone negatively influence upper limb motor recovery, as confirmed by the regression models. However, muscle tone is only a surrogate outcome of motor function, and it can of course be modified by rehabilitation, but not as much as other motor outcomes are, as confirmed by effect size of the two interventions. A description of the effect of intervention on muscle tone has been added by reporting effect size for each outcome measures in Table 2.

  1. Moreover, please analyze the flow diagram. It is not clear why each group of 62 patients was analyzed if several people in each group were either uninterrupted or/and did not continue to intervene.

Thanks to the reviewer for the suggestion. The authors apologies for uploading an incorrect version of the flow diagram, the graph has now been updated with the correct numbers. The description of how the patient were analyzed despite their leakage from the whole intervention is written according to the Intention To Treat method in the statistical analysis section (paragraph 2.6, lines 238 – 243).

Reviewer 3 Report

To check the number of participants in all paragraphs, e.g. line 207-208

Author Response

Response to Reviewer 3

Comments and suggestions for authors:

  1. To check the number of participants in all paragraphs, e.g. line 207-208

The calculated sample size of the trial was 122 participants (i.e. 61 for each group), as described in section 2.5. However, 124 patients were enrolled (i.e. 1 patient more for each group) because the patients eligible for that type of treatment had been screened and being only two who had completed the rehabilitation program more that those expected, the authors decided to keep them also for the analysis. The correct flow diagram has been uploaded, also taking into consideration the patients who were analyzed despite their leakage from the whole intervention, described according to the Intention To Treat method in the statistical analysis section (paragraph 2.6, lines 238 – 243).

Round 2

Reviewer 1 Report

Thanks to the authors for the modifications. However, most of the previous concerns were not addressed sufficiently. The manuscript cannot be recommended to be published in its current status. 

Author Response

The authors thank the Reviewer for new revision of the manuscript, which has been implemented according to Academic Editor suggestions. A more detailed description of tables has been added and the statement on Proportional Recovery Rule revised. The caption of each table has been improved with a summary description of tests performed, respective reasons and interpretations. Effect size calculation was modified with the calculation of Cohen’s d instead of biserial-rank correlation as recommended evidence cited in Gaskin et al. 2014 (ref. 28).
